# FGF diffusion is required for directed migration of postembryonic muscle progenitors in *C. elegans*

Theresa V. Gibney[1] and Ariel M. Pani[1,2,*]

## ABSTRACT

Extracellular signaling molecules mediate crucial aspects of cell–cell communication and play essential roles in development and homeostasis. Fibroblast growth factors (FGFs) are a family of secreted signaling proteins that can disperse long distances between cells and are often thought to form concentration gradients that encode spatial information. However, we know relatively little about the spatial distribution of FGFs *in vivo*, and endogenously tagged FGFs move between cells using different mechanisms in zebrafish and flies. We used FGF-dependent migration of *C. elegans* muscle progenitors called sex myoblasts (SMs) to elucidate FGF dispersal mechanisms and dissect how FGF guides migrating cells. Live imaging of cell dynamics and endogenously tagged FGF combined with membrane tethering and extracellular trapping approaches revealed that endogenous FGF is diffusible *in vivo* and extracellular dispersal is required for SM migration. Misexpression demonstrated that FGF is a bona fide chemoattractant that orients SMs during a critical window, while an unidentified, short-range signal acts in concert to position SMs precisely. Our finding that an invertebrate FGF is endogenously diffusible suggests that this may be the ancestral mode for FGF dispersal.

**KEY WORDS: Fibroblast growth factor, FGF, Diffusion, Gradient, Cell migration, Sex myoblast**

## INTRODUCTION

Cell–cell signaling and dynamic cellular interactions orchestrate animal development, homeostasis and regeneration. Cells use secreted signaling proteins to communicate at a range of distances to regulate patterning, growth, differentiation, and cell behaviors such as migration and morphogenesis (Wolpert, 2016; Muller and Schier, 2011). How secreted signaling proteins move between cells to reach their destinations *in vivo* remains largely unknown. Fibroblast growth factors (FGFs) are a family of secreted signaling proteins with essential roles in animal development (Dorey and Amaya, 2010; Thisse and Thisse, 2005), and abnormal FGF signaling is implicated in disease states (Turner and Grose, 2010; Babina and Turner, 2017). FGF proteins are often thought to form concentration gradients that encode spatial information and act as morphogens that regulate patterning and cellular behaviors over long distances (Dorey and Amaya, 2010; Muller et al., 2013; Thisse and Thisse, 2005; Balasubramanian and Zhang, 2016). Yet endogenous FGF protein gradients have only been observed in a handful of cases (Du et al., 2018; Harish et al., 2023; Toyoda et al., 2010; Dubrulle and Pourquie, 2004; Chen et al., 2009), and how endogenous FGF proteins move between cells *in vivo* has been debated based on data from a limited number of models. In vertebrates, endogenously tagged FGF8a forms long-range gradients by diffusion in developing zebrafish (Harish et al., 2023), and *Fgf8* mRNA decay coupled with growth contribute to an anteroposterior protein gradient in mouse and chick embryos (Dubrulle and Pourquie, 2004). However, two *Drosophila* FGFs are not freely diffusible and instead signal primarily at cell contacts (Stepanik et al., 2020; Du et al., 2018, 2022; Patel et al., 2022). In the absence of extracellular diffusion, FGFs can move between cells using dynamic cell membrane extensions known as cytonemes, which physically link signaling and responding cells (Du et al., 2018, 2022; Patel et al., 2022).

Major challenges for investigating how FGFs disperse in living animals include difficulties with visualizing endogenous signaling proteins *in vivo* along with challenges observing cytonemes, which are not preserved by standard fixation methods or visible using cytosolic fluorescent proteins (Sanders et al., 2013; Kornberg, 2017). Studies of FGF diffusion and gradient formation have often relied on exogenous fusion proteins, with the caveats that transgenes often do not recapitulate native expression patterns, and overexpression may alter extracellular dispersal dynamics and/or cell behaviors. Of the endogenously tagged FGF proteins examined so far, zebrafish FGF8a and *Drosophila* Branchless have been found to disperse between cells using different mechanisms (Du et al., 2018, 2022; Harish et al., 2023), and data from additional models are needed to discern underlying principles.

An ideal system to investigate FGF dispersal would allow for live imaging of FGF-secreting and -responding cells, visualization of endogenous FGF protein *in vivo*, and the ability to manipulate FGF dispersal without disrupting protein function or cell physiology. The migration of two *Caenorhabditis elegans* muscle progenitors called sex myoblasts (SMs) provides an experimentally tractable model to investigate FGF signaling mechanisms *in vivo*. All *C. elegans* postembryonic mesoderm cells, including SMs, are derived from the M mesoblast, which is located near the tail in early larvae. The M cell lineage gives rise to body wall muscles, coelomocytes, and two SMs that are located on the left and right sides of the animal (Sulston and Horvitz, 1977) (see Fig. 1). During the L2 larval stage, SMs migrate anteriorly to a precise position flanking the center of the gonad. SM migration requires the FGF8/17/18 family homolog EGL-17 (Burdine et al., 1997, 1998; Stern and Horvitz, 1991) and the FGF receptor (FGFR) EGL-15 (DeVore et al., 1995; Lo et al., 2008; Stern and Horvitz, 1991). SMs fail to complete their migration in *egl-17* and *egl-15* mutants (Stern and Horvitz, 1991), which leads to misplaced adult musculature and defects in egg

[1]Department of Biology, University of Virginia, Charlottesville, VA 22903, USA. [2]Department of Cell Biology, University of Virginia School of Medicine, Charlottesville, VA 22903, USA.

*Author for correspondence (amp2na@virginia.edu)

T.V.G., 0000-0001-5461-724X; A.M.P., 0000-0002-9338-9750

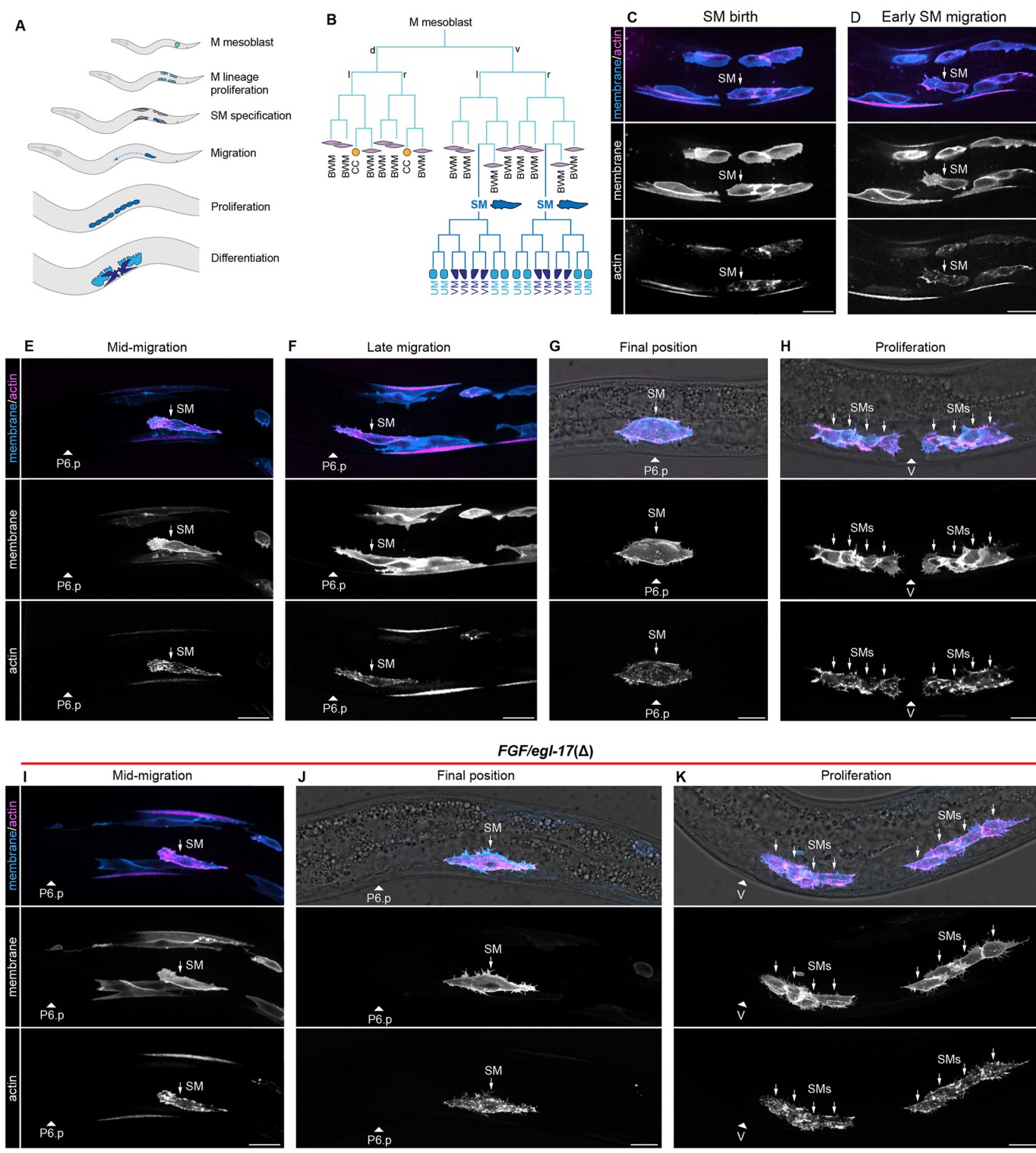

**Fig. 1. Live imaging of SM migration.** (A) Diagram of key developmental processes in the postembryonic mesoderm. (B) The M mesoblast gives rise to body wall muscles, coelomocytes, and sex myoblasts (SMs). SMs migrate anteriorly to flank the center of the gonad, proliferate, and differentiate into vulval and uterine muscles. (C-H) Live imaging of M lineage and SM dynamics using a single-copy *Phlh-8>2x mKate2::moesin ABD::F2A::2x mTurquoise2::PH* transgene to visualize plasma membranes (blue) and actin (magenta). (D) At the onset of SM migration, SMs delaminate from their sister body wall muscle cells and migrate anteriorly with broad, actin-rich protrusions and short filopodia. (E,F) SMs exhibit similar protrusions as they crawl towards and over the M-derived v17 and v18 body wall muscles. (G,H) At the end of their migration, SMs center over the uterine cells, dorsal to P6.p (G), before proliferating to form two groups of four cells flanking the vulva (H). (I-K) In *FGF/egl-17(Δ)* animals, SM migration (I) is initially indistinguishable from wild type, but SMs do not maintain polarized protrusions and fail to migrate past v17/18 (J). (K) SMs then proliferate in a posteriorly displaced location. Animals oriented with anterior to left and dorsal to top. Arrowheads denote position of P6.p or the vulva depending on stage. BWM, body wall muscle; CC, coelomocyte; SM, sex myoblast; UM, uterine muscle; V, vulva; VM, vulval muscle. Autofluorescent gut granules were removed in E-K by subtracting a 445ex/642-80em background channel (see Fig. S3). Images are representative of 15 or more samples. Scale bars: 10 µm.

laying. The prevailing model for SM migration is that FGF/EGL-17 is produced by the primary vulval precursor cell (VPC) P6.p and uterine cells and acts as a chemoattractant that guides myoblasts to their final positions (Branda and Stern, 2000; Burdine et al., 1998; Sherwood and Plastino, 2018).

Here, we used live imaging and genome engineering to characterize interactions between SMs and *egl-17*-expressing cells, visualize endogenous EGL-17 distribution, and test roles for FGF diffusion in SM migration. Live imaging revealed that SMs migrate near previously unknown FGF source cells but do not interact through cytonemes during migration. Intriguingly, endogenously tagged FGF did not form a visible protein gradient and localized primarily to FGF-expressing cells and SMs. Misexpression experiments showed that migrating SMs orient towards the strongest FGF source, but are only responsive to the direction of an FGF source during a critical time window. Anterior FGF misexpression also revealed somatic gonad cells can capture SMs migrating in close proximity, suggesting that an unidentified, short-range signal acts with directional information encoded by FGF to mediate precise SM positioning. Finally, extracellular protein trapping and an endogenously membrane-tethered EGL-17 demonstrated that endogenous FGF is diffusible, and its diffusion is required for SM migration.

## RESULTS
### SMs migrate in close proximity to *egl-17*-expressing cells but do not interact through cytonemes
Distinguishing between potential mechanisms for secreted protein dispersal between cells requires knowledge of cellular shapes and behaviors to assess the possibility of direct contact between cells. To investigate FGF dispersal mechanisms, we first characterized the architectures and behaviors of SMs throughout their migration. Although the *C. elegans* M lineage has been a model for cell fate decisions and migration over several decades (Chen and Stern, 1998; Branda and Stern, 2000; Sherwood and Plastino, 2018; Liu and Murray, 2023), previous studies mainly reported endpoint data, and high-resolution images of membrane and cytoskeletal architectures were not available. SMs are born in the L2 larval stage from an asymmetric division that gives rise to one SM and one body wall muscle (Sulston and Horvitz, 1977) (Fig. 1A,B). Using live imaging of plasma membrane and actin markers (Edwards et al., 1997), we observed that SMs delaminated from their sister body wall muscle cells and generated broad, actin-rich pseudopods polarized towards the anterior (Fig. 1C,D). SMs migrated anteriorly with broad, actin-rich protrusions at the leading edge of the cell along with short filopodia. SMs migrated anteriorly for ~20 μm before contacting, and then crawling over, the dorsal surfaces of the M lineage-derived ventral 17th and 18th body wall muscles (hereafter v17/18) (Fig. 1E,F), which are visible with the same transgene. After migrating over v17/18, SMs continued to produce broad protrusions at the leading edge as cells migrated over the surface of the somatic gonad, separated from underlying uterine cells by a basement membrane (Fig. S1). SMs then centered over the uterine progenitors (Fig. 1G), located dorsal to the primary VPC P6.p. After centering over the somatic gonad, each SM then underwent three rounds of division to generate two groups of four cells flanking the vulva (Fig. 1H). Throughout SM migration, we observed short filopodia, often near the leading edge of migrating cells, but we did not observe cytoneme-like structures linking migrating SMs to the P6.p and uterine cells thought to produce EGL-17. Intriguingly, SMs and their descendants did produce dynamic, cytoneme-like extensions after migration was complete (Movie 1). Observation of these structures at later developmental stages validated the ability of this transgene to

visualize cytonemes, and suggests that such protrusions are truly absent during SM migration rather than being present but undetectable with our methods.

As a baseline to characterize phenotypes caused by manipulating FGF, we first deleted the *egl-17* coding region. SM migration invariably failed in *FGF/egl-17(Δ)* animals and resembled *egl-17* and *egl-15(5a)* mutants. Live imaging showed that SM migration was unaffected until the point where SMs normally crawled over the v17/18 body wall muscles (Fig. 1I), which corresponds to the previously reported gonad- and FGF-independent early phase of SM migration (Branda and Stern, 2000). In *FGF/egl-17(Δ)* animals, after coming to rest atop the v17/18 body wall muscle cells, SMs failed to continue migrating (Fig. 1J) and no longer produced anteriorly polarized protrusions. SMs then proliferated in this posteriorly displaced location (Fig. 1K), leading to non-functional egg-laying muscles. To evaluate the possibility that differential capacity for FGF-Ras-ERK signaling underlies the transition to FGF-dependent migration, we examined ERK-nKTR biosensor (de la Cova et al., 2017) activity in SMs during the FGF-independent and FGF-dependent stages of migration. We found that ERK-nKTR activity was not significantly different between FGF-independent and FGF-dependent time points (Fig. S2), indicating that changes in ERK activity are unlikely to underlie differential functions for FGF during early and late SM migration.

After failing to observe cytoneme-like structures in migrating SMs, we turned to the identities and architectures of cells expressing *egl-17*, the FGF ligand required for SM migration. Transgenes driven by *egl-17* cis-regulatory sequences are expressed in the P6.p and dorsal uterine cells located at the endpoint of SM migration (Branda and Stern, 2000), which is key evidence supporting the model that EGL-17 acts as a chemoattractant to guide migrating SMs. However, transgenes often do not recapitulate endogenous expression, and the native expression pattern and dynamics for *egl-17* were not known. To visualize *egl-17*-expressing cells *in vivo*, we used Cas9-triggered homologous recombination to engineer an endogenous bicistronic gene encoding *egl-17* along with a plasma membrane marker by inserting *SL2::mNeonGreen(mNG)::PH* at the 3′ end of the endogenous *egl-17* coding sequence after the stop codon (Fig. 2B). *egl-17::SL2::mNG::PH* homozygotes were phenotypically indistinguishable from wild type and did not display SM migration defects. We used spinning disk confocal live imaging to characterize the *egl-17* expression pattern from the early L1 to L3 stages (Fig. 2A), focusing on the posterior half of the animal where the M lineage develops. To better visualize low levels of fluorescent protein expression in areas with autofluorescent gut granules, we removed broad-spectrum autofluorescence by subtracting a simultaneously acquired background channel (see Fig. S3). Before the first M cell division, we observed *egl-17::SL2::mNG::PH* expression in Q neuroblasts along with weaker expression in the V5 seam cells and a rectal cell. We also observed variable and faint fluorescence in the M cell itself (Fig. 2C). *egl-17::SL2::mNG::PH* expression then shifted near the time of the first M cell division, unexpectedly becoming highly expressed in ventral midline blast cells (Fig. 2D) along with the M lineage cells themselves (Fig. 2D,E). As animals continued to develop, *egl-17::SL2::mNG::PH* expression diminished in M lineage cells but was maintained at high levels in a continuous row of cells along the ventral midline (Fig. 2F,G; Fig. S4). After the birth of the SMs, we no longer observed *egl-17::SL2::mNG::PH* expression in the M lineage. During the early phase of SM migration, *egl-17::SL2:: mNG::PH* remained highly expressed in the P4.p-P8.p blast cells, with lower expression in their neuronal progeny (Fig. 2H). We also observed low, but consistent expression in V5-derived neuroblasts

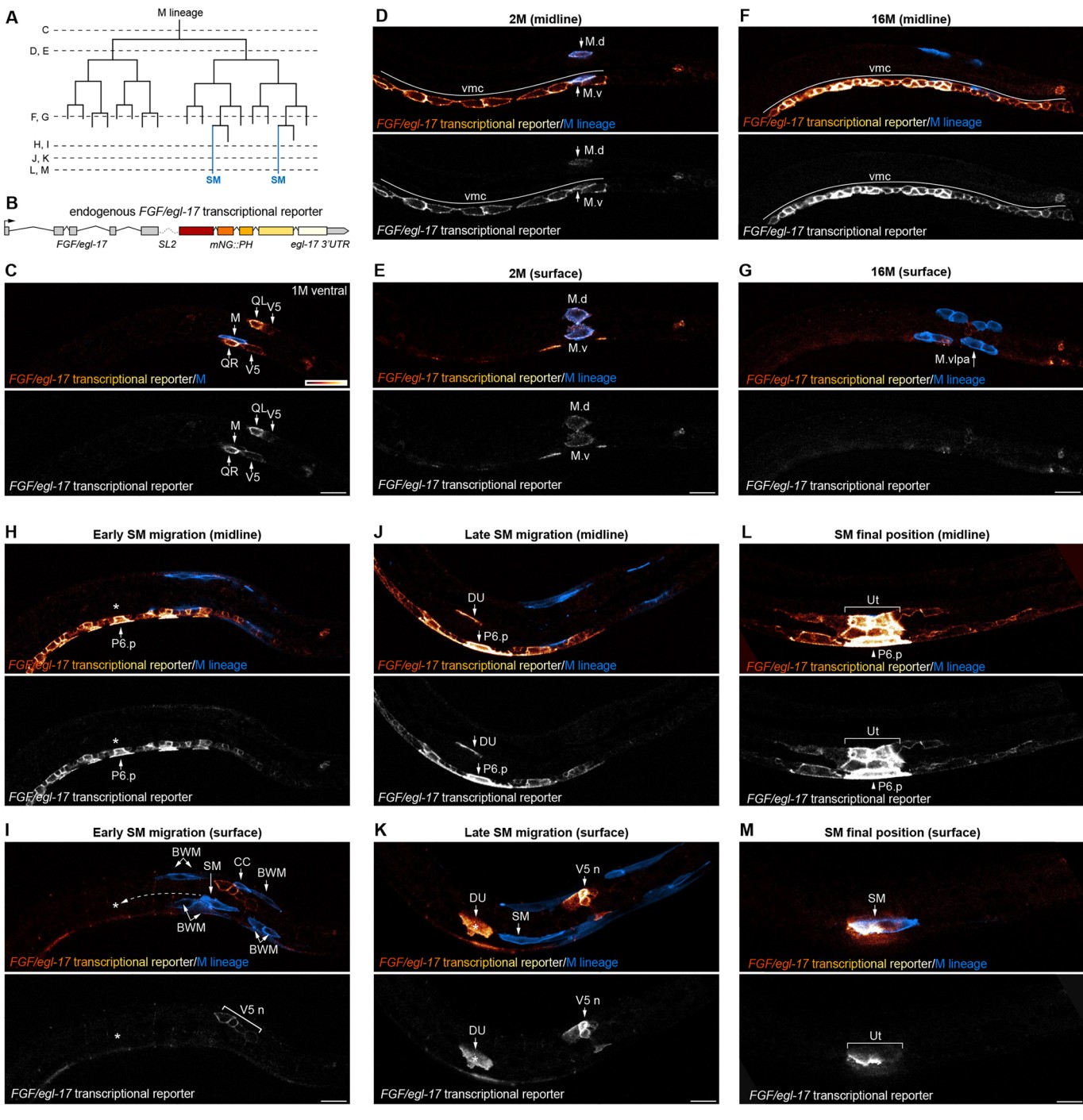

**Fig. 2. *In vivo* imaging of an endogenous *egl-17* transcriptional reporter revealing dynamic expression and multiple EGL-17 sources.**
(A) Schematic of M lineage divisions. Dashed lines indicate time points for the images in C-M. (B) Design of the bicistronic, endogenous *egl-17* transcriptional reporter expressing *egl-17* and the membrane marker mNG::PH under native regulatory control. (C) Ventral view of an early L1-stage animal showing *egl-17* expression in Q neuroblasts along with weak expression in V5 seam cells, a rectal cell, and M. (D,E) Lateral midline (D) and surface (E) views of a later L1-stage animal showing *egl-17* expression in ventral midline blast cells and the M lineage. (F,G) Midline (F) and surface (G) views at the 16M stage showing strong *egl-17* expression in a row of ventral midline cells. The M.vlpa and M.vrpa cells divide to generate one SM and one additional body wall muscle on each side of the animal. (H,I) Midline (H) and surface (I) views showing *egl-17* expression in ventral midline cells, including neurons and presumptive VPCs, during early SM migration. Asterisks denote the SM migration endpoint, and dashed arrow indicates the approximate extent of the EGL-17-dependent migration phase. *egl-17* is also expressed in V5-derived neuroblasts dorsal to the position of the SM. (J,K) Midline (J) and surface (K) views showing *egl-17* expression in the dorsal uterine cell, VPCs, and V5-derived neurons during a late stage of SM migration. See Movie 2 and Fig. S5 for time-lapse imaging at this stage. (L,M) Midline (L) and surface (M) views showing strong *egl-17* expression in dorsal uterine cells and P6.p at the end of SM migration, along with weaker expression in other VPCs and somatic gonad cells. Animal in C is in ventral view with anterior to left. Animals in D-M are in lateral views oriented with anterior to left and dorsal to top. Autofluorescent gut granules were removed from images by subtracting a 445ex/642-80em background channel. BWM, body wall muscle; CC, coelomocyte; DU, dorsal uterine cell; SM, sex myoblast; Ut, uterine cells; V5n, V5-derived neuroblasts or neurons; vmc, ventral midline cells. Images are representative of 15 or more samples. Scale bars: 10 μm (bar in C applies to C-G; bar in I applies to H-M).

located on the dorsal side of the animal, slightly posterior to the concurrent position of migrating SMs (Fig. 2I). As SMs migrated over the M-lineage-derived v17/18 body wall muscles and approached the somatic gonad, *egl-17::SL2::mNG::PH* was upregulated in dorsal uterine cells and P6.p but downregulated in other ventral midline cells (Fig. 2J). However, strong expression was maintained in V5-derived neurons (Fig. 2K). SMs crawled forwards using a broad protrusion at the leading edge and short filopodia as they migrated towards the *egl-17*-expressing dorsal uterine cell (Movie 2; Fig. S5). As SMs migrated over the somatic gonad and centered over the *egl-17*-expressing uterine progenitors (Fig. S6), *egl-17::SL2::mNG::PH* remained highly expressed in the dorsal uterine and P6.p cells with lower expression in other uterine cells, P5.p, P7.p, and other ventral midline cells (Fig. 2L,M). We did not observe cytoneme-like protrusions from *egl-17::SL2::mNG::PH*-expressing cells, although SMs migrated in close proximity to *egl-17*-expressing midline cells and neurons.

### EGL-17 orients migrating SMs

The finding that *egl-17* was expressed in the early M lineage and then in ventral midline cells along the path taken by migrating SMs raised the possibility that *egl-17* may act permissively in at least some aspects of SM migration. Given the available tools, earlier studies concluded that EGL-17 acts as a chemoattractant based on transgene expression patterns (Branda and Stern, 2000; Burdine et al., 1998) combined with the ability of SMs to follow the *egl-17*-expressing primary VPCs (Burdine et al., 1998) or the somatic gonad (Thomas et al., 1990) when they were misplaced. *egl-17* is required for primary VPCs to mediate SM positioning in the absence of the somatic gonad (Burdine et al., 1998), but the use of genetic backgrounds including an overexpressed *Ras/let-60(G13E)* gain-of-function allele and/or laser ablation of the somatic gonad or VPCs complicates our understanding of the normal role for EGL-17 in guiding SM migration. Our results showing that *egl-17* is strongly expressed in ventral midline cells before VPC induction and in V5-derived neuroblasts at the end of SM migration raise the possibility that EGL-17 is required permissively for SMs to migrate anteriorly and SM positioning depends on a complementary 'stop here' signal in wild-type animals. To distinguish between instructive and permissive roles for FGF, we used single-copy transgenes to drive *egl-17* expression in M lineage cells, at uniform levels along the anteroposterior axis, or in tail cells in *FGF/egl-17(Δ)* (Fig. 3) and wild-type (Fig. S7) animals.

We reasoned that if FGF acts permissively, then expressing *egl-17* cell-autonomously or at even levels throughout the body should rescue SM migration in *FGF/egl-17(Δ)* animals (Fig. 3A,B). We first used an *hlh-8* promoter to drive *FGF/egl-17::mNG* expression in M lineage cells including the SMs (Fig. 3C). *Phlh-8>FGF/egl-17::mNG* failed to rescue SM migration (n=114/114) in *FGF/egl-17(Δ)* animals (Fig. 3C), indicating that cell-autonomous FGF expression is not sufficient for SM migration. As a complementary test for permissive signaling, we used the *myo-3* promoter to express *FGF/egl-17::mNG* at uniform levels along the anteroposterior body axis. *Pmyo-3>FGF/egl-17::mNG* expression similarly failed to rescue SM migration (n=68/68) (Fig. 3D). To test the extent to which FGF acts as a chemoattractant for migrating SMs, we then used a fragment of the *egl-20* promoter to drive *FGF/egl-17::mNG* in tail cells in an attempt to reorient migrating SMs towards the posterior. In *FGF/egl-17(Δ)* animals, SMs typically arrested their migration over the v17/18 body wall muscle cells (Fig. 3B). Expressing *FGF/egl-17::mNG* in the tail led to posteriorly displaced SMs (n=173/174) that were located near the most posterior M-derived body wall muscles (Fig. 3E). Because

this position is near the birthplace of the SMs (see Fig. 1), we used live imaging at earlier time points to assess whether the initial FGF-independent phase of migration failed. Intriguingly, we observed that initial SM migration occurred normally, and cells generated oriented protrusions and migrated anteriorly until they reached v17/18 (Fig. 3F). However, SMs failed to maintain their polarity and produced misoriented protrusions. SMs then moved posteriorly, although with a less polarized morphology compared to normal migration (Fig. 3F). Taken together, these results demonstrate that FGF acts instructively to orient SMs during the FGF-dependent phase of migration and that SMs are insensitive to an ectopic FGF source until the point when their migration normally requires FGF.

To test the ability of ectopic FGF sources to interfere with normal SM migration, we performed complementary misexpression experiments in a wild-type background. Cell-autonomous EGL-17 expression strongly disrupted SM migration (n=75/80), leading to under-migration reminiscent of FGF loss of function (Fig. S7). Uniformly expressing EGL-17 in body wall muscle cells led to similar under-migration defects in the vast majority (n=102/104) of cells (Fig. S7). High levels of EGL-17 expression in the tail disrupted SM migration in most animals (n=35/58), resulting in cells that migrated towards the anterior but did not complete their migration (n=9/58) and cells that migrated towards the posterior (n=26/58) (Fig. S7). Together, these findings confirm that EGL-17 acts instructively to orient migrating SMs.

### *egl-17(Δ)* somatic gonad cells can mediate precise SM positioning

To explore further the ability of EGL-17 to attract migrating SMs, we used mosaic *Pmyo-3>FGF/egl-17::mNG* expression to generate animals with ectopic point sources of EGL-17 in varying positions in the *FGF/egl-17(Δ)* background. In many cases, SMs centered over ectopic EGL-17-producing body wall muscles that were in the posterior half of the animal (Fig. 3G). However, in rare cases in which mosaic EGL-17-secreting cells were in a similar anteroposterior position as the somatic gonad (n=4), we observed that SMs extended towards, or centered over, the somatic gonad rather than the ectopic FGF-expressing cell(s) (Fig. 4A). To determine the extent to which somatic gonad cells are capable of capturing SMs migrating towards an ectopic FGF source, we expressed *egl-17* in the pharynx in an *FGF/egl-17(Δ)* background using a single-copy transgene driven by a *myo-2* promoter (Fig. 4B). Anterior *egl-17* misexpression led to complex SM migration phenotypes, including over-migration (42% of cells), normal positioning over the dorsal uterine cells and P6.p (39%), and under-migration (19%) (Fig. 4B-E). The observation of under-migrated cells suggests that SMs may be born near the tail end of a potential FGF gradient in these animals where FGF levels may be near a lower limit required for migration guidance. SMs that stopped at their normal position maintained that position throughout proliferation and differentiation (Fig. S8). These results suggest that *FGF/egl-17(Δ)* somatic gonad cells provide a yet-to-be-identified, short-range cue that can over-ride conflicting positional information from an ectopic FGF source. The ability of SMs to migrate in these animals also indicates that *egl-17* expression in the early M-lineage is not strictly required for SM migration.

### Endogenous EGL-17 does not form a visible protein gradient *in vivo*

Understanding how secreted signaling proteins disperse requires the ability to visualize endogenous proteins in their native tissue context. Although FGF gradients are hypothesized to regulate many developmental processes, endogenous FGF protein gradients have

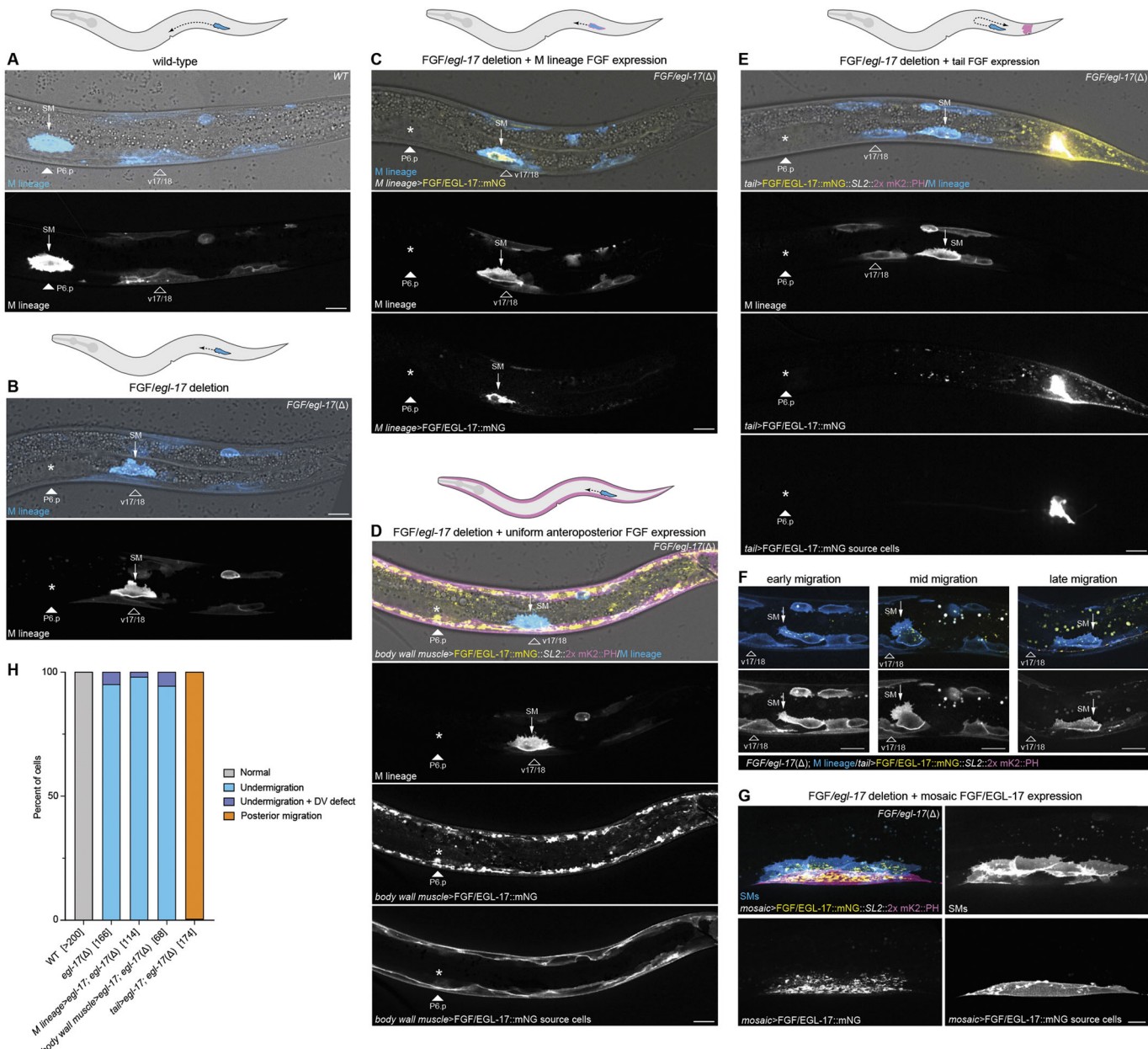

**Fig. 3. EGL-17 acts as an instructive signal to orient migrating SMs.** (A) Final SM position over the uterine cells and P6.p in a wild-type animal. (B) Final SM positioning in an *FGF/egl-17(Δ)* animal showing migration arrest over the v17/18 body wall muscles. Asterisk marks the normal migration endpoint. (C) Expressing FGF/EGL-17::mNG in M lineage cells using the *hlh-8* promoter does not rescue SM migration in an *FGF/egl-17(Δ)* background. (D) Uniform expression of FGF/EGL-17::mNG along the anteroposterior axis using the *myo-3* promoter does not rescue SM migration in an *FGF/egl-17(Δ)* background. (E) Expressing FGF/EGL-17::mNG in tail cells in an *FGF/egl-17(Δ)* background leads to more posterior SM positioning. (F) SMs initially migrate anteriorly in *Pegl-20⁻¹²⁶¹⁻⁶¹⁰>FGF/egl-17::mNG; FGF/egl-17(Δ)* animals but fail to maintain their polarity and then move towards the posterior. Tail cells expressing FGF/EGL-17::mNG are located out of the field of view to the right. (G) Isolated body wall muscle cells expressing FGF/EGL-17::mNG in mosaic animals can position SMs in the *FGF/egl-17(Δ)* background. (H) Summary of SM positioning phenotypes. Bracketed numbers indicate the number of cells scored for each genotype with two cells per animal (left and right). See Table S1 for source data. Asterisks denote the normal migration endpoint in images of cells with migration defects. 2x mKate2::PH marks membranes of FGF-expressing cells in D-H. All images oriented with anterior to left and dorsal to top. SM, sex myoblast; WT, wild type. Autofluorescent gut granules were removed in B-E by subtracting a 445ex/642-80em background channel. Scale bars: 10 μm.

only been directly observed in a handful of cases. To visualize EGL-17 protein in living animals, we endogenously tagged EGL-17 at its C terminus with mNG::3xFlag (hereafter EGL-17::mNG). Knock-in animals were phenotypically indistinguishable from wild type, and FGF/EGL-17::mNG was visible at low levels throughout SM migration using spinning disk live imaging. We primarily observed FGF/EGL-17::mNG protein localized to cell types that expressed

the *FGF/egl-17::SL2::mNG::PH* transcriptional reporter (see Fig. 2) and in M lineage cells, along with isolated punctae associated with other cell types (Fig. 5A-F). During the FGF-independent phase of SM migration, we observed FGF/EGL-17::mNG internalized within SMs and other M lineage cells (Fig. 5B). As SMs continued to migrate, we observed increasing accumulation of intracellular FGF/EGL-17::mNG in SMs with little visible

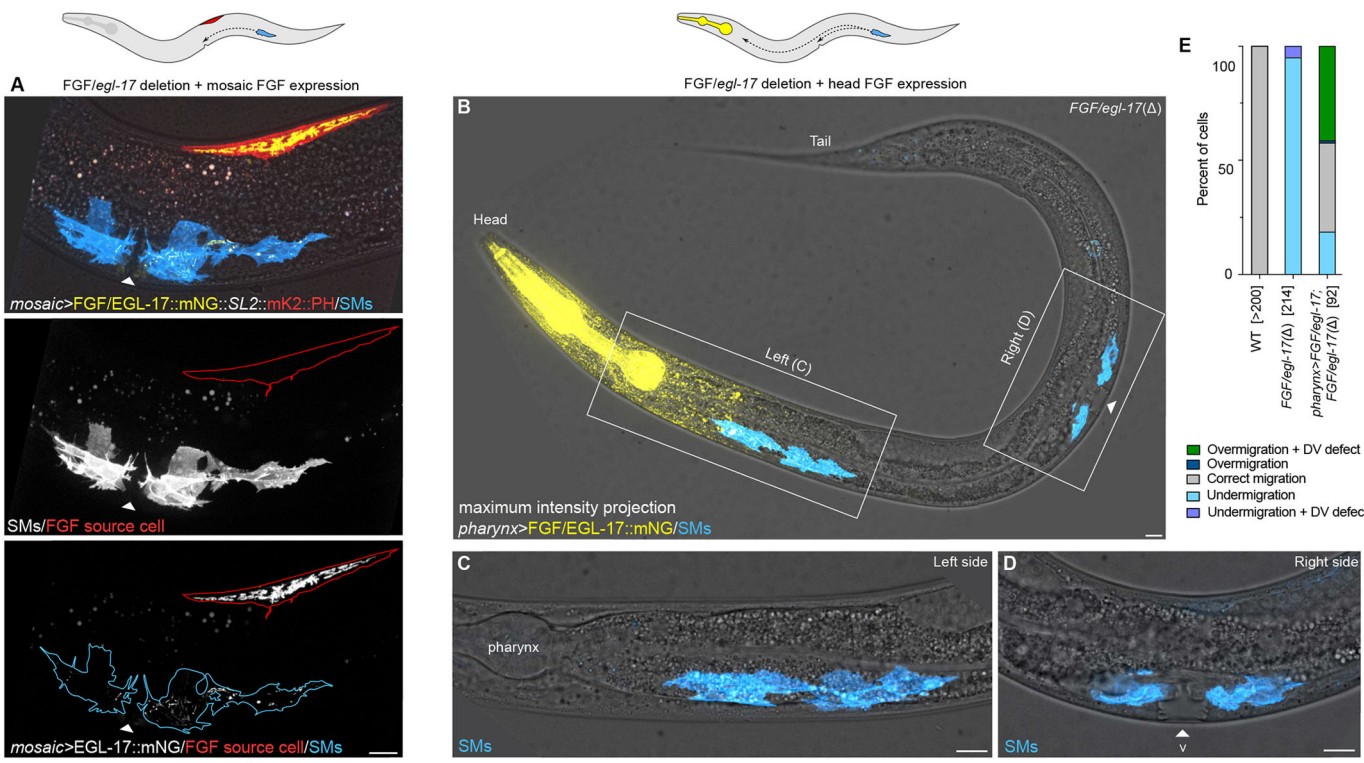

**Fig. 4. FGF/egl-17(Δ) gonad cells can capture SMs migrating towards a misplaced EGL-17 source.** (A) Proper SM descendant positioning in an animal with mosaic *Pmyo-3>FGF/EGL-17::mNG* expression in a dorsal body wall muscle near the normal migration endpoint. The SM descendants are located in their normal position on the ventral side flanking the somatic gonad center rather than near the FGF/EGL-17::mNG-expressing cell, marked by co-expressed 2x mKate2::PH. (B) Misexpressing FGF/EGL-17::mNG in the head in an *FGF/egl-17(Δ)* background attracts migrating SMs, but *FGF/egl-17(Δ)* somatic gonad cells can also capture migrating SMs and mediate normal positioning. B shows a maximum intensity projection through an entire *Pmyo-2>FGF/EGL-17:: mNG*; *FGF/egl-17(Δ)* animal where the left SM over-migrated, and the right SM was normally positioned. White triangle marks the normal SM migration endpoint. (C,D). Higher magnification views of over-migrated (C) and normally positioned (D) SM descendants in the same animal. (E) Summary of SM positioning phenotypes in wild-type (WT), *FGF/egl-17(Δ)* and *Pmyo-2>FGF/EGL-17::mNG*; *FGF/egl-17(Δ)* animals. Numbers in brackets indicate the number of cells scored for each genotype with two cells per animal (left and right). See Table S1 for source data. All images are oriented with anterior to left and dorsal to top. Autofluorescent gut granules were removed in B-D by subtracting a 445ex/642-80em background channel. Scale bars: 10 µm.

fluorescence in additional cell types (Fig. 5C-F). We used fluorescence recovery after photobleaching (FRAP) to test the extent to which EGL-17 within migrating SMs was protein retained from pre-migratory stages (Fig. 5G). We photobleached SMs at the onset of the FGF-dependent phase of migration, recovered animals to allow SMs to migrate, and imaged them again near the end of migration. EGL-17 fluorescence recovered in all cells examined (*n*=10/10), which indicates that SMs take up EGL-17 during their migration.

Unexpectedly, FGF/EGL-17::mNG protein did not form a visible, extracellular concentration gradient at any stage. Because FGF/EGL-17::mNG fluorescence was only slightly higher than background, we used autofluorescence subtraction (Fig. S3) to visualize background-free localization of FGF/EGL-17::mNG at the onset of the FGF-dependent stage of SM migration. Maximum intensity projections of background-subtracted images showed FGF/EGL-17:: mNG localization within the dorsal uterine cell, P6.p, ventral midline cells and V5-derived neuroblasts (Fig. 5D). Aside from *egl-17*-expressing cells, we observed bright punctae within migrating SMs and isolated punctae associated with other cell types (Fig. 5D). Even with autofluorescence subtraction, we did not observe an FGF/EGL-17::mNG gradient or protein associated with basement membranes or extracellular space. However, it is possible that FGF is present in a gradient at levels below the detection limit for *in vivo* spinning disk confocal imaging. Although the absence of an obvious EGL-17 protein gradient was unanticipated, this result matches expectations

for a freely diffusing protein in the absence of a uniformly expressed extracellular binding partner.

**EGL-17 diffusion is required for SM migration**

To elucidate potential requirements for EGL-17 diffusion, we sought to prevent extracellular dispersal of EGL-17 without disrupting contact-dependent signaling. We first engineered an endogenously membrane-anchored EGL-17 by inserting mNG fused to a NLG-1 transmembrane domain (Wang et al., 2014; Teichmann and Shen, 2011) at the 3′ end of *egl-17* (Fig. 6A,B). *FGF/egl-17::mNG::nlg-1*$^{TM}$ animals displayed fully penetrant defects in SM migration (*n*=162/ 162; Fig. 6B,D), demonstrating that endogenously membrane-anchored EGL-17 is incapable of supporting normal development. To confirm that membrane-anchored FGF/EGL-17::mNG::NLG-1$^{TM}$ remained capable of contact-dependent signaling, we used the *myo-3* promoter to express *FGF/egl-17::mNG::nlg-1*$^{TM}$ in body wall muscles, which directly contact SMs. *Pmyo-3>FGF/egl-17::mNG:: nlg-1*$^{TM}$ animals exhibited SM migration defects (*n*=98/98; Fig. S9) similar to the phenotype caused by untethered EGL-17 misexpression in body wall muscles (Fig. S7; Table S1), thereby validating that FGF/ EGL-17::mNG::NLG-1$^{TM}$ was competent to activate signaling at cell contacts.

As a parallel approach to test the extent to which EGL-17 diffusion is required for SM migration, we sought to limit extracellular EGL-17 dispersal without functionally modifying the protein itself. We first attempted to sequester extracellular EGL-17

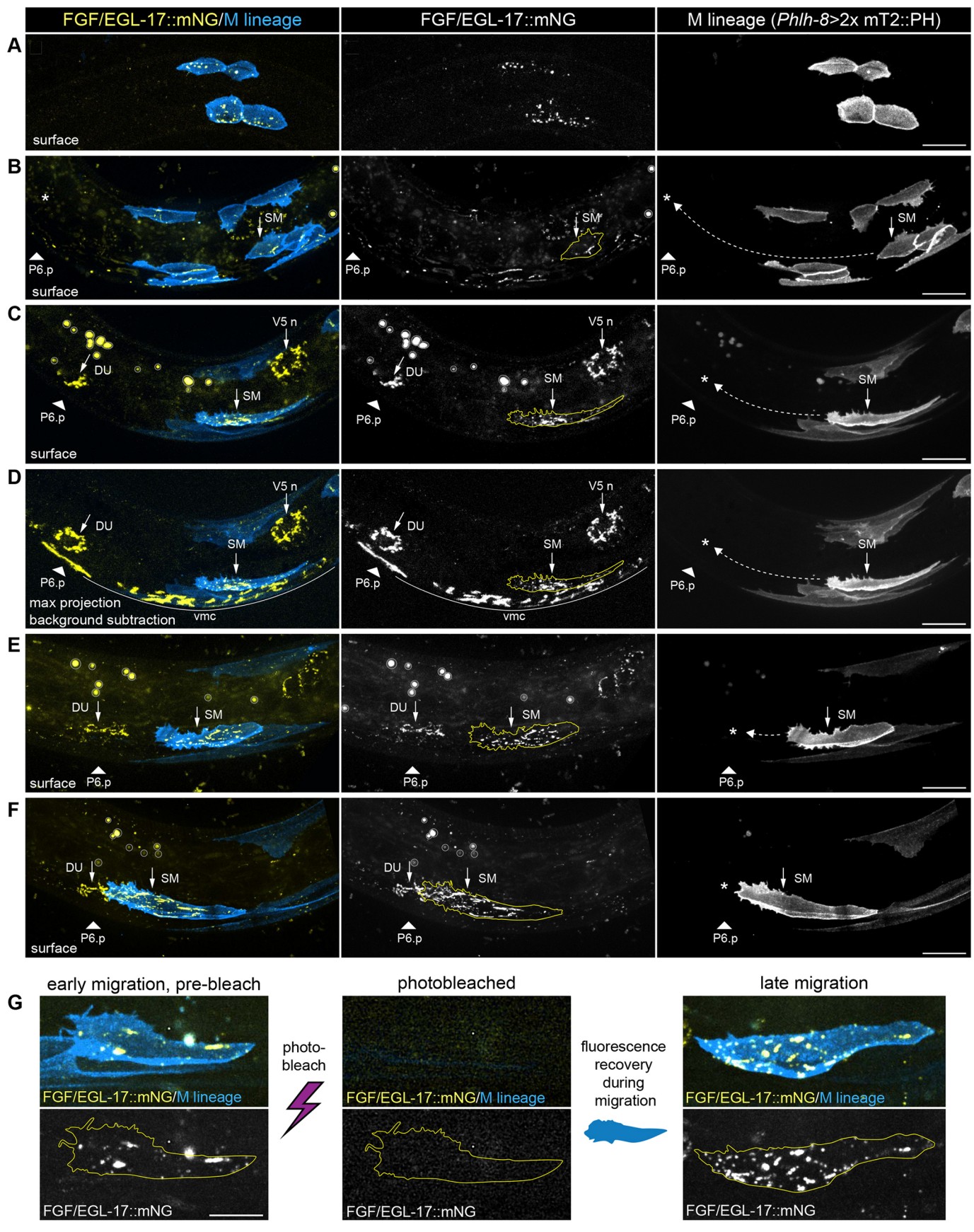

**Fig. 5.** See next page for legend.

**Fig. 5. *In vivo* visualization of endogenously tagged EGL-17::mNG during SM migration.** (A) Surface view showing that FGF/EGL-17::mNG localizes to intracellular punctae in M lineage cells prior to SM birth. (B) Surface view showing that, shortly after birth of the SMs, FGF/EGL-17::mNG localizes primarily to M lineage cells with sparse signal associated with other cell types. Dashed arrow indicates the path of SM migration, and asterisk indicates the endpoint. (C) Surface view showing FGF/EGL-17::mNG localization in the plane of SM migration at the beginning of the FGF-dependent migration. (D) Maximum intensity projection of background-subtracted images showing FGF/EGL-17::mNG localization in the same animal as in C. Strong, intracellular FGF/EGL-17::mNG signal is visible in the SM along with the *egl-17*-expressing dorsal uterine cell, P6.p and other ventral midline cells, and V5-derived neuroblasts. Note the absence of an observable protein gradient between the migrating SM and its destination and only sparse FGF/EGL-17::mNG localization to other cell types. (E) Surface view showing FGF/EGL-17::mNG localization in the plane of SM migration prior to contact between the SM and dorsal uterine cell. (F) Surface view showing FGF/EGL-17::mNG localization in the plane of SM migration after contact between the SM and dorsal uterine cell. (G) Photobleaching followed by fluorescence recovery during migration demonstrates SMs internalize FGF/EGL-17::mNG while migrating. All images are oriented with anterior to left and dorsal to top. Asterisks denote the presumptive endpoint of SM migration. Autofluorescent gut granules appear as circular artifacts in gut cells that are larger than FGF/EGL-17::mNG punctae and are outlined in gray in C, E and F. DU, dorsal uterine cell; SM, sex myoblast; V5 n, V5-derived neuroblasts or neurons; vmc, ventral midline cells. Images are representative of 20 or more samples. Scale bars: 10 μm.

using an anti-GFP nanobody-based approach known as Morphotrap (Harmansa et al., 2015), similar to that used to test roles for FGF8a diffusion in zebrafish embryos (Harish et al., 2023). We tagged endogenous EGL-17 with the GFP-derivative YPET and crossed EGL-17::YPET animals to a Morphotrap strain that was previously used to demonstrate a requirement for WNT/EGL-20 diffusion (Pani and Goldstein, 2018). However, Morphotrap was unable to capture FGF/EGL-17::YPET, which did not localize to Morphotrap-expressing body wall muscle cells (Fig. S10). As expected given its failure to visibly sequester FGF/EGL-17::YPET, *Pmyo-3>Morphotrap* had no effect on SM migration (Fig. S10; Table S1; *n*=76/76).

As an alternative approach to capture the diffusing fraction of EGL-17 protein, we used a single-copy transgene to express *egl-15(5a)* along with a *2x mKate2::PH* membrane marker in body wall muscles to provide a receptor-mediated sink for FGF/EGL-17::mNG (Fig. 6C). We did not observe defects in body wall muscle differentiation or anatomy. While endogenously tagged FGF/EGL-17::mNG does not localize to body wall muscles in wild-type animals (see Fig. 5C-F), we observed conspicuous FGF/EGL-17::mNG localization to body wall muscles near *FGF/egl-17*-expressing cells in *FGF/egl-17::mNG; Pmyo-3>FGFR/egl-15(5a)::SL2::2x mKate2::PH* animals (Fig. 6C; Fig. S11). Consistent with the ability of this transgene to capture FGF/EGL-17::mNG, we observed migration defects in the majority of SMs (*n*=101/194; Fig. 6C,D). Like *egl-17* deletion phenotypes, we primarily observed under-migration along with rarer dorsoventral defects (Fig. 6D; Fig. S11; Table S1). Because the *egl-15(5a)*-misexpressing body wall muscles are not directly juxtaposed between the SMs and *egl-17*-expressing cells, we hypothesized that *egl-15(5a)* misexpression interfered with migration by sequestering EGL-17 from a common pool of extracellularly diffusing protein, thereby reducing the amount of available FGF below the level required for SM migration. To assess this model, we used spinning disk live imaging with a photon number-resolving qCMOS camera to quantify the amount of FGF/EGL-17::mNG internalized by SMs in wild-type and *Pmyo-3>FGFR/egl-15(5a)* animals (Fig. 6E-I). To avoid potentially confounding effects of cell position, we only utilized cells that migrated to their normal position for comparisons. Even when SMs

migrated to their normal positions, we found that *Pmyo-3>FGFR/egl-15(5a)* significantly reduced the amount of FGF/EGL-17::mNG internalized by SMs at both the one SM (Fig. 6E-F′,I) and eight SM stages (Fig. 6G-I) (Mann–Whitney test *P*<0.0001 at both stages). This result confirmed that SMs acquire EGL-17 from a general pool of extracellularly mobile protein even though SMs at the one SM stage directly overlie *egl-17*-expressing cells (see Fig. S6).

## DISCUSSION

*In vivo* visualization of endogenous EGL-17, experimental manipulations of FGF dispersal, and visualization of signaling and responding cell dynamics provide direct evidence that EGL-17 is natively diffusible and that free, extracellular FGF dispersal is required for SM migration. While the FGF signaling pathway is evolutionarily conserved, analyses of endogenously tagged FGF proteins have shown disparate mechanisms for FGF movement between cells and gradient formation in vertebrates (Harish et al., 2023; Dubrulle and Pourquie, 2004; Yu et al., 2009) and *Drosophila* (Du et al., 2018, 2022; Stepanik et al., 2020). In developing zebrafish, FGF8a diffuses extracellularly to form long-range protein gradients (Harish et al., 2023), and multiple studies have demonstrated FGF diffusion in vertebrate models using overexpression-based approaches (Toyoda et al., 2010; Yu et al., 2009; Duchesne et al., 2012). However, the *Drosophila* FGFs characterized to date do not freely spread between cells by diffusion but instead signal at cell contacts (Du et al., 2018, 2022; Stepanik et al., 2020) and form gradients through a cytoneme-based mechanism in cases where gradients exist (Du et al., 2018). Our results demonstrating that *C. elegans* EGL-17 moves between cells by diffusion, and that diffusion is required for at least one of its key functions, suggest that some invertebrate FGFs are diffusible and that this dispersal mechanism is a common feature of FGF proteins across phyla. Interestingly, the *Drosophila* FGFs that signal at cell contacts include extended C-terminal domains with a transmembrane domain (Stepanik et al., 2020) or GPI anchor (Du et al., 2022) that are absent in *C. elegans* EGL-17 and vertebrate FGF8/17/18 family proteins. However, the *C. elegans* FGF9/16/20 homolog LET-756 also includes a large C-terminal domain, and studies investigating FGF/LET-756 signaling mechanisms may provide additional insights into the diversity of mechanisms used for FGF dispersal.

While our findings demonstrate that EGL-17 signaling in SM migration relies on diffusion, they do not rule out roles for cytonemes in other aspects of SM development or FGF/EGL-17 signaling in other contexts. Indeed, the observation that stationary SMs extend numerous filopodia after migration has ended and throughout proliferation and morphogenesis could be consistent with a role for cytonemes in communication between SMs and other cell types at later developmental stages. Modes of signaling protein dispersal may also vary depending on tissue architectures (Stapornwongkul and Vincent, 2021) that affect the abilities of signaling and responding cells to make direct contacts. During SM migration, somatic gonad cells are encased within a basement membrane that would interfere with cytoneme-based signaling between the EGL-17-expressing uterine cells and migrating SMs. However, the fact that FGF8a moves by extracellular diffusion in early zebrafish development (Harish et al., 2023) demonstrates that diffusion can be the predominant mode for FGF dispersal even when there are not physical barriers to cell contacts.

SMs are a long-standing paradigm to study directed cell migration (Chen and Stern, 1998; Sherwood and Plastino, 2018), but technical limitations prevented early studies (Burdine et al., 1998) from directly testing whether EGL-17 functions as a chemoattractant that guides

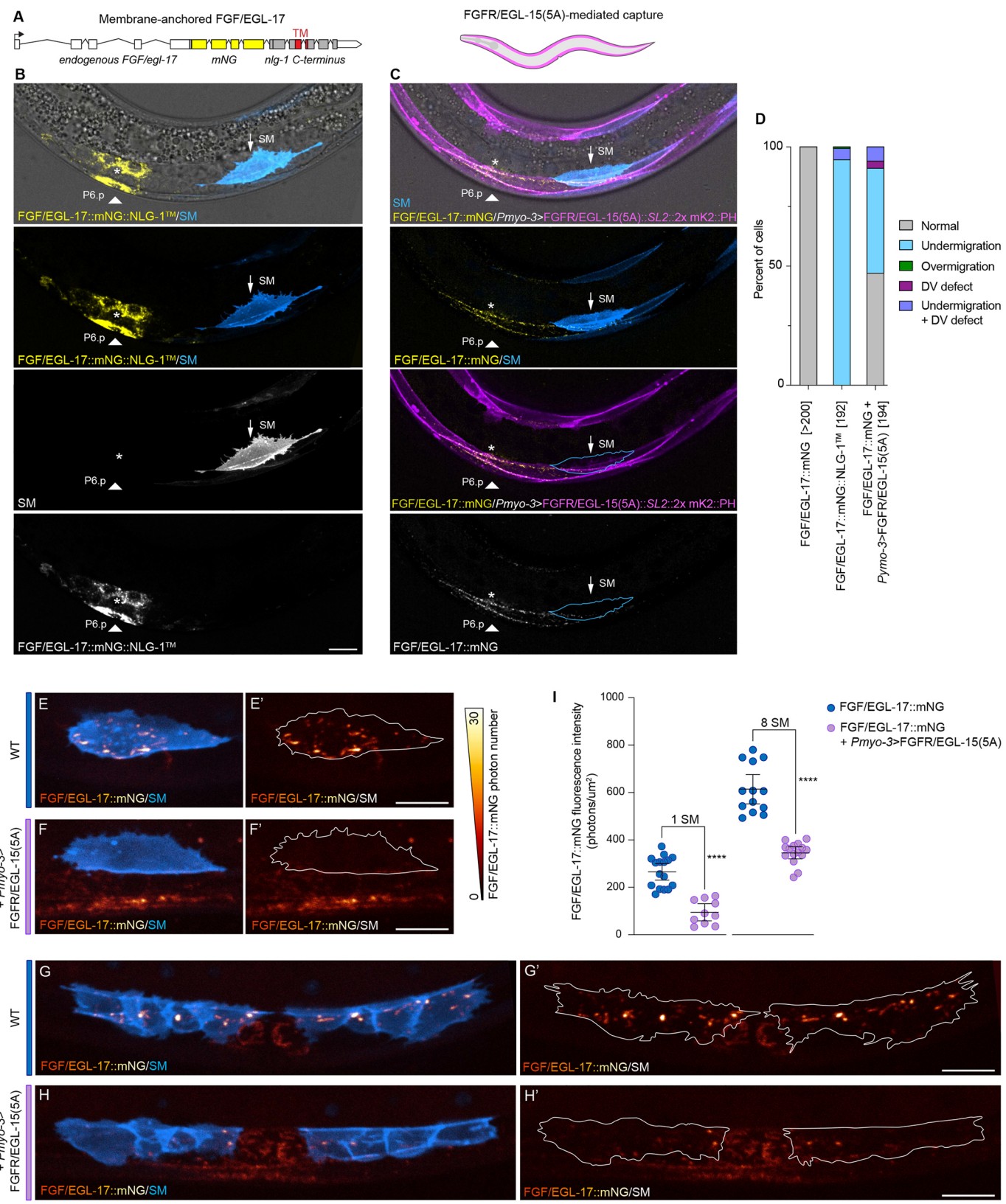

**Fig. 6.** See next page for legend.

migrating SMs towards the highest source of FGF or as a permissive signal that allows cells to migrate anteriorly until receiving a second cue from the somatic gonad or primary VPCs. Our *egl-17* misexpression results demonstrate that elements of both models are

correct. EGL-17 is not a permissive signal for SM migration as neither cell-autonomous nor spatially uniform *egl-17* expression can rescue SM migration in *FGF/egl-17(Δ)* animals, and both manipulations disrupt SM migration in a wild-type background.

**Fig. 6. EGL-17 diffusion is required for SM migration.** (A) Schematic of the membrane-tethered *egl-17* allele. Endogenous *egl-17* coding and 3′UTR sequences are indicated in white, *mNG* is shown in yellow, and the *nlg-1* C-terminus is shown in gray with the transmembrane (TM) domain indicated in red. (B) FGF/EGL-17::mNG::NLG-1^TM is not capable of supporting SM migration. White triangle indicates P6.p and asterisk indicates the normal endpoint of SM migration. (C) Expressing FGFR/EGL-15(5A) in body wall muscles sequesters endogenous FGF/EGL-17::mNG and disrupts SM migration. Note that endogenous FGF/EGL-17::mNG localizes to body wall muscles near *egl-17*-expressing cells in *Pmyo-3>FGFR/EGL-15(5A):: SL2::2x mKate2::PH; FGF/egl-17::mNG* animals but not the SM. White triangle indicates P6.p and asterisk indicates the normal endpoint of SM migration. (D) Summary of SM positioning phenotypes in *FGF/egl-17::mNG, FGF/egl-17::mNG::nlg-1^TM* and *Pmyo-3>FGFR/EGL-15(5A)::SL2::2x mKate2::PH* animals. Numbers in brackets indicate the number of cells scored for each genotype with two cells per animal (left and right). See Table S1 for source data. (E-F′) Comparison of FGF/EGL-17::mNG localization in SMs at the end of migration in an *FGF/egl-17::mNG* animal (E,E′) and a *Pmyo-3>FGFR/EGL-15(5A)::SL2::2x mKate2::PH; FGF/egl-17:: mNG* with normal SM migration (F,F′). EGL-17 captured by body wall muscles is visible ventral to the SM. (G-H′) Comparison of FGF/EGL-17:: mNG localization in SMs at the eight SM stage in an *FGF/egl-17::mNG* animal (G,G′) and a *Pmyo-3>FGFR/EGL-15(5A)::SL2::2x mKate2::PH; FGF/egl-17::mNG* animal with normal SM migration (H,H′). (I) Quantification of FGF/EGL-17::mNG levels in SMs in control and *Pmyo-3>FGFR/EGL-15(5A)::SL2::2x mKate2::PH* animals with normal migration. Expressing FGFR/EGL-15(5A) in body wall muscles significantly reduces FGF/EGL-17:: mNG even in SMs that correctly migrated (Mann–Whitney test, ****$P<0.0001$ at both time points). Data are mean with 95% confidence intervals. All images are oriented with anterior to left and dorsal to top. Images in E-H and images used for quantification in I were acquired using a Hamamatsu ORCA Quest quantitative CMOS camera in its photon number resolving mode. Autofluorescent gut granules were removed in B and C by subtracting a 445ex/642-80em background channel. Scale bars: 10 μm.

Misexpressing *egl-17* in tail cells reversed the direction of SM migration, demonstrating that extracellular FGF can orient migrating SMs towards a distant source. However, sparse mosaic and pharyngeal *egl-17* expression also highlighted the ability of *FGF/ egl-17(Δ)* somatic gonad cells to attract SMs over short distances, or to halt their migration when they are in close proximity. The molecules that mediate these interactions remain to be identified. It is possible that the proteins that mediate short-range communication between the gonad and SMs are related to the gonad-dependent repulsive guidance signal that repels migrating SMs in the absence of EGL-17 or EGL-15(5A) (Branda and Stern, 2000; Sherwood and Plastino, 2018), which also awaits discovery.

*C. elegans* has a single FGFR homolog, *egl-15*, the extracellular domain of which is alternatively spliced to generate receptor isoforms with differential affinity for EGL-17 and the second *C. elegans* FGF ligand, LET-756 (Lo et al., 2008; Goodman et al., 2003). Our finding that expressing the EGL-15(5A) isoform in body wall muscle leads to SM migration defects by sequestering extracellular EGL-17 suggests a novel rationale for multiple FGFR isoforms in *C. elegans*. LET-756 signaling in the hypodermis is required for larval viability, which is mediated by EGL-15(5B). Although EGL-15(5A) is capable of performing the essential functions of EGL-15(5B) (Lo et al., 2008), expressing levels of EGL-15(5A) sufficient for LET-756 signaling in the hypodermis could interfere with SM migration given that EGL-17 is produced at very low levels. In this case, alternative FGFR isoforms may have evolved in part to protect a low-expressed chemoattractant, EGL-17, from being functionally sequestered by FGFR-expressing cells that respond to the more abundant LET-756. It is possible that avoiding the ability to sequester related ligands and prevent their dispersal to target cells is an underappreciated function for receptor diversification.

FGFs can act as morphogens (Toyoda et al., 2010; Harish et al., 2023; Kengaku and Okamoto, 1995), and FGF gradients are thought to be important for conveying positional information. In classic models, spatial gradients of FGF protein are essential to activate distinct concentration (or fold change)-dependent responses in cells located at varying distances from an FGF source (Balasubramanian and Zhang, 2016). While we demonstrated that EGL-17 acts instructively to orient mesodermal progenitor migration in *C. elegans*, endogenous EGL-17 unexpectedly did not form a visible protein gradient *in vivo* during SM migration. Nonetheless, cell-autonomous, uniform and posterior EGL-17 misexpression experiments demonstrated that a localized EGL-17 source is required for SM migration, and migrating cells move towards the strongest FGF source. While we were unable to directly observe an endogenous, extracellular EGL-17 gradient, our experimental results could be explained by a subtle gradient below the detection limit of our current *in vivo* microscopy methods. The presence of FGF/EGL-17:: mNG gradients in misexpression experiments also indicates that this protein is capable of diffusing to form visible gradients under some conditions. The fact that substantially decreased levels of EGL-17 are still capable of mediating SM migration in some *Pmyo-3>FGFR/egl-15(5a)* animals highlights that extremely small quantities of EGL-17 are sufficient for this biological function. Our finding that endogenous EGL-17 does not form an obvious gradient may also be explained by the fact that freely diffusible proteins require extracellular binding partners to form stable, long-range concentration gradients. Theoretical predictions and experimental tests using secreted GFP demonstrated that freely diffusible proteins do not form concentration gradients on their own *in vivo* (Stapornwongkul et al., 2020), and the length and shape of a secreted protein gradient depends on the distribution of binding partners and their affinities (Stapornwongkul et al., 2020; Stapornwongkul and Vincent, 2021; Muller et al., 2013). Interestingly, the FGFR isoform EGL-15(5A), which binds EGL-17 and is required for SM migration, is expressed predominantly in the M lineage (Lo et al., 2008), suggesting that other nearby cell types may lack high-affinity binding partners. It is possible that the absence of an observable, endogenous EGL-17 gradient reflects a lack of extracellular binding partners expressed in cells outside the M lineage. Secreted protein gradients can also arise through binding to non-receptor binding partners with lower affinities (Stapornwongkul and Vincent, 2021; Stapornwongkul et al., 2020). In many contexts, FGFs associate with heparan sulfate proteoglycans in the extracellular matrix (Duchesne et al., 2012; Muller et al., 2013; Balasubramanian and Zhang, 2016), but such interactions are not known for EGL-17, and viable *C. elegans* heparan sulfate proteoglycan mutations have not been reported to affect SM migration. Collectively migrating cells can generate their own gradients in migrating tissues (Dona et al., 2013; Venkiteswaran et al., 2013), but such a mechanism has not been described for individually migrating cells *in vivo*. Our work demonstrates that invertebrate FGFs may have been ancestrally diffusible and also lays the groundwork for investigations of how migrating cells interpret multiple types of extracellular signals to navigate to precise destinations *in vivo*.

## MATERIALS AND METHODS

### *C. elegans* maintenance

*C. elegans* animals were cultured on Nematode Growth Medium (NGM) plates, fed *Escherichia coli* OP50, and maintained at 20°C for experiments. Only well-fed animals were used for live imaging and/or scoring SM positioning.

### Plasmid construction

Homologous repair templates for the *egl-17* locus were constructed by inserting homology arm PCR products into plasmids containing a

self-excising selection cassette using New England Biolabs HiFi DNA assembly master mix as described in detail elsewhere (Dickinson et al., 2015). To construct the repair template used to delete the *egl-17* gene, we constructed an mNG^SEC^3xFlag repair template with homology arms that flanked the coding region to replace the entire *egl-17* coding sequence with mNG. To generate the repair template for the *egl-17* transcriptional reporter, we cloned *SL2::mNG::PH* in place of *mTurquoise2* in pDD315 and used the same homology arms as for endogenous protein tagging. A stop codon was added between the *egl-17* coding sequence and *SL2*. To generate the repair template for membrane-tethered FGF/EGL-17::mNG::NLG-1$^{TM}$, we used the pDD268 backbone and the same homology arms as for FGF/EGL-17:: mNG with the addition of a 2265 bp genomic fragment encoding the last 184 amino acids of NLG-1 between 3xFlag and the stop codon and 3′ homology arm. The NLG-1 C terminus includes a single transmembrane domain and was previously used as a C-terminal membrane tether for functional, membrane-anchored Netrin/UNC-6 (Teichmann and Shen, 2011; Wang et al., 2014). Plasmid backbones for single copy insertions near the ttTi4348 and ttTi5605 sites were described previously (Pani and Goldstein, 2018). The plasmid backbone for single-copy insertions at Chr IV:4,237,723 was constructed by cloning flanking genomic homology arms with the guide RNA target sequence deleted into pDD315 with the *mTurquoise2* and *3xHA* sequences removed. Coding sequences for transgenes were codon-optimized using the *C. elegans* codon adapter tool (Redemann et al., 2011). Transgene promoters, fluorescent proteins, and the *SL2* sequence were amplified from genomic DNA or existing plasmids and inserted into linearized backbones using New England Biolabs HiFi DNA assembly. The *egl-20*$^{(−1261–610)}$ enhancer was cloned upstream of a *pes-10* minimal promoter. To construct plasmids for Cas9+sgRNA expression, we cloned guide RNA sequences into the *Peft-3>Cas9+sgRNA* expression vector pDD162 using site-directed mutagenesis. Guide RNA target sites used were (5′-3′): *egl-17* N terminus, TCGACAACATCAGGGTGAGT; *egl-17* C terminus, CACATGATAG-TTTGTATCGT; Chr I transgenes, GAAATCGCCGACTTGCGAGG; Chr II transgenes, GATATCAGTCTGTTTCGTAA; Chr IV transgenes, ACTGT-TGGATGCCTGTGTAG. Sequences for the *nlg-1* C-terminal region and transgene promoters used are provided in supplementary Materials and Methods.

### Transgenesis and genome engineering

All strains generated here were made using Cas9-triggered homologous recombination with a self-excising selection cassette (SEC) (Dickinson et al., 2015). Single-copy transgenes were inserted into chromosomal safe harbor locations near the ttTi4348 site on Chr I, the ttTi5605 site on Chr II, or a new site at Chr IV:4,237,723. EGL-17 was endogenously tagged at the C terminus to generate mNG, YPET, and mNG::NLG-1$^{TM}$ fusion alleles. The endogenous *egl-17* transcriptional reporter was engineered by inserting *SL2::mNG::PH* immediately after the stop codon using the same guide RNA. For transgenesis and endogenous tagging, we co-injected the homologous repair template, Cas9+sgRNA plasmid, and extrachromosomal array markers into adult germlines as described previously (Dickinson et al., 2013, 2015). We handled animals with a fine paintbrush during the microinjection process to increase survival and throughput (Gibney et al., 2023). Candidate knock-in animals were selected using hygromycin B, a dominant roller phenotype, and the absence of fluorescent extrachromosomal array markers. To isolate homozygous lines, we picked single animals to new NGM plates without hygromycin B and screened for animals with 100% roller progeny. To excise the selectable marker cassette, we heat-shocked homozygous, young L1 larvae for 4 h at 34°C to induce *heat shock>Cre* expression and picked non-roller offspring in the next generation. To generate strain APL670 for Morphotrap experiments, we crossed FGF/EGL-17::YPET animals with an existing *Pmyo-3>Morphotrap^SEC* strain (Pani and Goldstein, 2018) followed by heat shock to excise the SEC. To visualize ERK activity, we crossed ljfSi41; ljfSi42 animals to an existing *Phlh-8>ERK-nKTR* strain (de la Cova et al., 2017).

### Microscopy and image processing

Larval animals were immobilized using 0.1 mmol/l levamisole in M9 buffer (Chai et al., 2012) for standard live imaging or 0.1 μm polystyrene nanoparticles without anesthetic (Kim et al., 2013) for time-lapse imaging

of cell migration. Animals were mounted between a high-precision cover glass (Thorlabs, CG15CH) and 3-5% (wt/vol) agarose pads. Animals were imaged within 1 h of mounting, and images shown are representative of at least ten animals. Numerous animals were mounted on each slide and selected for imaging based on developmental stage and orientation. Images were acquired using a Yokogawa CSU-X1 spinning disk confocal with a Hamamatsu ORCA Fusion BT sCMOS camera or a Yokogawa CSU-W1 SoRa spinning disk confocal with a Yokogawa Uniformizer and Hamamatsu ORCA Quest qCMOS camera. Both confocal units were mounted on Nikon Ti2 inverted microscope stands. Imaging was performed using 445 nm, 514 nm, 561 nm or 594 nm lasers for excitation and 480/40, 545/40, 575LP, 610LP or 642/80 emission filters depending on fluorescent protein. Imaging was performed using Nikon Plan Apo IR 60×/1.27 NA water immersion, Apo TIRF 60×/1.49 NA oil immersion, or Apo TIRF 100×/1.49 NA oil immersion objectives. Images were acquired using Nikon NIS Elements Advanced Research software (5.42.04 and 5.42.06) with camera exposure times and laser intensities adjusted based on experimental requirements and fluorescence intensities of the imaged proteins. For animals in which the areas of interest were not located within a single field of view, we acquired tiled images using the large image acquisition mode with automated image stitching. Stitched images were not used for fluorescence intensity measurements. Images were deconvolved using Nikon NIS Elements software. Brightness, contrast and colors were adjusted using Fiji (Schindelin et al., 2012), and figures were prepared using Adobe Illustrator 28.3.

*C. elegans* larvae and adults possess gut granules with broad-spectrum autofluorescence that can interfere with visualization of fluorescent proteins by live imaging. Gut granule autofluorescence is correlated across excitation and emission wavelengths (Rodrigues et al., 2022) and is prominent with all laser excitation wavelengths and emission filters used in this study. To better visualize cell architectures and weakly expressed endogenous proteins, we subtracted gut granule autofluorescence in some images (see Fig. S3) using a method similar to SAIBR (Rodrigues et al., 2022). To subtract autofluorescence, we first acquired background images using 445 nm excitation paired with 642/80 or 575LP emission filters, which allowed us to spectrally separate broad-spectrum autofluorescence from the fluorescent proteins used here. We adjusted excitation power such that gut granules in the background channel were slightly more intense than gut granules in the fluorescent protein channels (445ex, 480/40em; 514ex, 545/40em; 561ex, 575LPem; 594ex, 642/80em) and then acquired paired background and fluorescent protein images at each *z*-position. We then subtracted the background channel from the fluorescent protein channel(s) using the 'subtract' option in the Fiji Image Calculator (see Fig. S3). In cases in which gut granules in the background channel were substantially brighter than in a fluorescent protein channel, we subtracted a constant from all pixels in the background channel to roughly equalize gut granule fluorescence intensity across channels before subtracting the background channel.

FRAP was performed using a 405 nm laser and Acal-BFi UV-Optimicroscan photomanipulation device operated by NIS Elements software. For FGF/EGL-17::mNG FRAP experiments, we mounted multiple, synchronized larvae at the onset of the FGF-dependent stage of SM migration. We then photobleached the SMs in all animals on the slide, recovered them from the slide by removing the cover glass and rinsing with M9 buffer and allowed animals to recover on NGM plates with *E. coli* OP50. After 4 h, we re-mounted the previously photobleached animals on slides for spinning disk imaging.

### Quantitative analyses

We scored larval animals for SM migration defects using *Phlh-8*-driven transgenes to visualize SMs at developmental stages when they have normally completed their migration. At the one SM stage, we used somatic gonad development and/or the number of VPCs for staging as the SMs finish their migration prior to the first dorsal uterine cell and VPC divisions. At the one SM stage, we scored SMs as normally migrating when the single SM was centered over the uterine cells and P6.p. For animals imaged at later developmental stages, we scored SMs as normally migrating when the SM progeny were centered over the somatic gonad and symmetrically located flanking the P6.p descendants or the vulva. Cells that migrated anterior to the dorsal uterine cells or P6.p were scored as over-migrating. Cells that were

located posterior to P6.p but anterior to the most posterior M-derived body wall muscles were scored as under-migrating. Cells located over, or posterior to, the most posterior M-derived body wall muscles were scored as posteriorly migrating. To quantify FGF/EGL-17::mNG levels, we imaged animals using a Hamamatsu ORCA Quest qCMOS camera in its photon number resolving mode for absolute measurements of fluorescence intensity. To calculate FGF/EGL-17::mNG photon number, we drew a region of interest using the *Phlh-8>2x mTurquoise2::PH* membrane marker and measured FGF/EGL-17::mNG photon number in a *z*-projection of image planes encompassing the SM. Statistical significance of the difference in fluorescence intensity between wild-type and *Pmyo-3>FGFR/egl-15(5a)::SL2::2x mKate2::PH* backgrounds was assessed using a Mann–Whitney test. To quantify ERK-nKTR activity, we calculated the ratio of ERK-nKTR nuclear to cytoplasmic fluorescence intensity using co-expressed mCherry::H2B to visualize the nucleus (de la Cova et al., 2017). For ERK-nKTR measurements, we subtracted off-worm background in a nearby region from the raw pixel intensities. We assessed statistical significance using a Kolmogorov–Smirnov test. All statistical tests were performed using GraphPad Prism 10.2.2 software. No statistical tests were used to predetermine sample size. Animals from at least two plates were imaged on separate days for all strains except for mosaic misexpression of *Pmyo-2>egl-17::mNG::sl2::2x mKate2::PH*. No animals or data points were excluded post hoc.

### Acknowledgements
Some strains were provided by the CGC, which is funded by the National Institutes of Health Office of Research Infrastructure Programs [P40 OD010440]. pRCA5 was a gift of Rebecca Adikes.

### Competing interests
The authors declare no competing or financial interests.

### Author contributions
Conceptualization: T.V.G., A.M.P.; Formal analysis: T.V.G., A.M.P.; Funding acquisition: T.V.G., A.M.P.; Investigation: T.V.G., A.M.P.; Methodology: T.V.G., A.M.P.; Project administration: A.M.P.; Supervision: A.M.P.; Visualization: T.V.G., A.M.P.; Writing – original draft: T.V.G., A.M.P.; Writing – review & editing: A.M.P.

### Funding
This work was supported by the National Institutes of Health (R35GM142880 to A.M.P.; Predoctoral Fellowship F31HD112152 to T.V.G.). Open Access funding provided by National Institutes of Health. Deposited in PMC for immediate release.

### Data and resource availability
*C. elegans* strains generated in this study have been deposited at the *Caenorhabditis* Genetics Center (CGC) (https://cgc.umn.edu/). Plasmids and other reagents will be shared on request. All other relevant data and details of resources can be found within the article and its supplementary information.

### Peer review history
The peer review history is available online at https://journals.biologists.com/dev/lookup/doi/10.1242/dev.204802.reviewer-comments.pdf

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
