## [Peer Review File · Development (Cambridge, England)]

FGF diffusion is required for directed migration of postembryonic muscle progenitors in *C. elegans*

Theresa V. Gibney and Ariel M. Pani

DOI: 10.1242/dev.204802

Editor: Swathi Arur

Review timeline

Original submission: 24 March 2025

Editorial decision: 25 April 2025

First revision received: 30 July 2025

Accepted: 13 August 2025

Original submission

First decision letter

MS ID#: dev.204802

MS Title: FGF diffusion is required for directed migration of postembryonic muscle progenitors in *C. elegans*

Authors: Theresa V Gibney; Ariel M Pani

Article Type: Research Article

Dear Dr Pani,

I have now received all the referees' reports on the above manuscript, and have reached a decision. The referees' comments are appended below, or you can access them online: please go to:

As you will see, the referees express considerable interest in your work, but provide recommendations to significantly improve the manuscript, both textually and for rigor. If you are able to revise the manuscript along the lines suggested, which may involve further experiments, I will be happy receive a revised version of the manuscript. Your revised paper will be re-reviewed by one or more of the original referees, and acceptance of your manuscript will depend on your addressing satisfactorily the reviewers' major concerns. Please also note that Development will normally permit only one round of major revision.

Please attend to all of the reviewers' comments and ensure that you clearly highlight all changes made in the revised manuscript. Please avoid using 'Tracked changes' in Word files as these are lost in PDF conversion. I should be grateful if you would also provide a point-by-point response detailing how you have dealt with the points raised by the reviewers in the 'Response to Reviewers' box. If you do not agree with any of their criticisms or suggestions please explain clearly why this is so.

Reviewer 1

SUMMARY OF THE ADVANCE MADE IN THIS PAPER AND ITS POTENTIAL SIGNIFICANCE TO THE FIELD

The manuscript dev.204802, titled "FGF diffusion is required for directed migration of postembryonic muscle progenitors in *C. elegans*," dissects how FGF guides migrating cells. Using

live microscopy combined with endogenously tagged FGF and extracellular trapping and membrane tethering, the authors demonstrate that FGF is diffusible, and that extracellular dispersal is required for sex myoblast migration. The methods used in this paper addresses a challenge that exists for studying how FGFs disperse in living organisms. Prior to the methods used in this paper, it was difficult to visualize endogenous signaling protein gradients *in vivo* in addition to observing cytonemes. The authors use FGF signaling (dispersal) to demonstrate their novel methods in conjunction with established CRISPR/Cas9 gene-editing techniques to visualize *in vivo* signaling protein gradients. More specifically, the authors utilize the migration of two *C. elegans* muscle progenitor cells (sex myoblasts, SMs) to demonstrate the power of their methods while at the same time, increasing our understanding of FGF mediated sex myoblast migration.

Using live imaging, the authors provided a look into the architecture and behaviors of SMs during their migration. These studies provide novel data as previous data has largely focused on “endpoint data” as high-resolution images of membrane and cytoskeletal architectures were not available. Authors document SM migration and observe short filopodia (near the leading edge of migrating cells) but did not observe cytoneme-like structures.

The authors also deleted the entire FGF/*egl-17* coding region and characterized SM migration and actin dynamics. Live imaging showed that without the *egl-17* coding region, SMs failed to continue migrating past v17/18 body wall muscle positions demonstrating that there is an FGF independent and FGF dependent migration phase. Using ERK-nKTR biosensor methods, the authors demonstrated that differences in ERK signaling are unlikely the cause of the differential functions for FGF signaling during SM migration. Using CRISPR/Cas9 methods, the authors created endogenous tagged strains that allowed them to demonstrate that while there were no cytoneme-like protrusions from *egl-17* expressing cells, SMs were in close proximity to *egl-17* expressing midline cells and neurons during much of their migration.

The use of the *egl-17* transcriptional reporter allowed the authors to uncover a more dynamic expression pattern than previous possible with other published transgenes. Their studies were able to confirm that EGL-17 acts instructively to guide migrating SMs and also suggests that these cells are able to differentiate the relative strengths of opposing EGL-17 sources. Examination of worms with ectopic point sources of EGL-17 the authors then demonstrated that *egl-17(D)* somatic gonad cells provide a yet to be identified, short-range cue that has the ability to override conflicting positional information from ectopic FGF sources.

Finally using meticulous microscopy and CRISPR/Cas9 methods, the authors showed that while EGL-17 does not appear to form a visible protein gradient *in vivo*, diffusion of EGL-17 is required for SM migration.

Taken together, using novel microscopy methods with amazing images, the author provide data that contributes to our current understanding of EGL-17 mediates sex myoblast migration. Given the conserved nature of the FGF signaling pathway, these data can provide insights into both non-*C. elegans* FGF signaling and even more broadly, our understanding of the diversity of secreted signaling molecules and their mechanisms for function.

Overall, this manuscript is well-written, and the experiments described are meticulously executed. I only have minor comments on how the manuscript can be improved.

SUGGESTIONS TO AUTHORS Minor comments:

line 52: It might be worthwhile to mention the mutant phenotype when SM migration is disrupted.

Figure 1A: The detail in the “differentiation” worm schematic is difficult to see.
Figure 1B: UM and VM are hard to distinguish due to the size of the font, however, the colors of the cells help.

Figure 2: If possible, increase the size of the font used to label the cells.

line 234: It would be helpful to define “normal positioning.” Is this the precise center of the gonad (if so, how is this determined)? Or is there a small range relative to ventral cell positions?

line 266: The authors state that there was the absence of a clear protein gradient observed, is it possible that there is a subtle gradient that is undetectable with the current technology / methods? If possible, please address this possibility.

Figure S7: The fonts in both A and B are difficult to read, please consider increasing the font size. Statistical analyses would enhance the data interpretation.

line 513: Does the type of high-precision cover glass matter? If so, can details be provided?

line 554: Include criteria for identification as L3 and L4 stages.

Reviewer 2

SUMMARY OF THE ADVANCE MADE IN THIS PAPER AND ITS POTENTIAL SIGNIFICANCE TO THE FIELD

This work by Gibney and Pani addresses the in vivo behavior of an FGF ligand in controlling sex myoblast migration in larval *C. elegans*. Using genome editing, microscopy and a set of tools developed in other work, this work demonstrates that FGF discretely control aspects of cell polarization and migration and that diffusible FGF is required. Ectopic expression of FGF can interfere with the direction of polarization and migration. They conclude that FGF is a bona fide chemoattractant in vivo that acts over a distant and can direct migration. Authors show that FGF is a diffusible attractant for cell migration in vivo with an instructive role. A strength is that the work directly addresses questions of FGF signaling in an in vivo system, questions that can be muddled by misleading results when an in vitro approach is employed.

SUGGESTIONS TO AUTHORS

There are a lot of data associated with this work, so much so that at times it was difficult to keep it all straight and to ensure a critical eye on every single piece of data presented. The genome editing strategies are clever, and the imaging very convincing. The work is rigorous and generally the proper controls are included. A strength is that the work directly addresses questions of FGF signaling in an in vivo system, questions that can be muddled by misleading results when an in vitro approach is employed.

Comments:

Are cytonemes "seeable" in *C. elegans*? A reference of an example of visualizing *C. elegans* cytonemes would help. Otherwise, it is difficult to rule out cytonemes.

In Figure 3G with ectopic *egl-17* expression, the cell is clearly not as polarized to the anterior as wild type. However, it is not polarized to the posterior, which is described in the text. It would be helpful to show a posteriorly polarized cell in this figure. demonstrating instructive reorientation by a diffusible FGF cue.

While this work clearly shows an instructive role for diffusible FGF, there is no evidence of any sort of functional gradient.

p. 8 The authors mention "the cells are capable of measuring the relative strengths of opposing sources" of FGF. What is the basis for measuring relative strength of opposing sources to correlate with the phenotypic effect? I think the authors are implying that these cells can sense different amounts of FGF and respond differently. However, the cells might simply be attracted to the strongest source, and not comparing levels.

Reviewer 3

SUMMARY OF THE ADVANCE MADE IN THIS PAPER AND ITS POTENTIAL SIGNIFICANCE TO THE FIELD

In this manuscript, Gibney and Pani used a combination of live imaging, endogenously tagged FGF, membrane tethering and extracellular trapping to determine how FGF guides the migration of the *C. elegans* sex myoblasts (SMs), which give rise to the hermaphrodite egg-laying muscles. While it has been previously well established that FGF signaling is critical for SM migration, the live imaging experiments allowed the authors to assemble a nice visual catalog of the SM migration process. Furthermore, the authors found that dispersal of endogenous FGF/EGL-17 is important for SM migration. They also obtained additional evidence supporting the previous finding that FGF/EGL-17 functions as a chemoattractant to orient SMs. The manuscript is well written. The strengths of the paper are the quality of the live imaging data and the *in vivo* evidence using endogenously tagged FGF/EGL-17. It will be of interests to a wide range of developmental biologists interested in cell-cell signaling, in particular FGF signaling.

SUGGESTIONS TO AUTHORS

I have some concerns regarding the interpretation of some of the experimental data.

- 1) The authors stated that the endogenously tagged EGL-17::mNG is very faint. Thus it is possible that endogenous EGL-17 has a graded distribution. However, the lower levels of EGL-17 might be under the limit of detection. In fact, the image shown in Figure 4B using the *myo-2* promoter to drive EGL-17::mNG clearly showed a graded distribution of the mNG signal away from the pharynx where EGL-17::mNG is produced.
- 2) The expression of EGL-17 in the M lineage cells is intriguing. While the authors showed that expressing EGL-17 in the M lineage is not able to rescue the SM migration phenotype of *egl-17* mutants, is M lineage expression of EGL-17 required for SM migration? In other words, does M lineage depletion of EGL-17 affect SM migration?
- 3) It is not hard to imagine why the authors observed both normal migration as well as under migration and posterior migration when they expressed ELG-17 in the tail in WT animals. However, in *Pmyo-2* > FGF/EGL-17::mNG; FGF/*egl-17(0)* animals, a substantial number of animals have under-migrated SMs (Figure 4E). The authors should comment on this.
- 4) Please comment on the internalized EGL-17::mNG in migrating SMs. Are these perdurance of EGL-17::mNG that was initially expressed in the M lineage cells before SMs were born, or are these internalized EGL-17::mNG taken up by the SMs along the migrating path?

First revision

Author response to reviewers' comments

Responses to the Reviewers

We thank the Reviewers for their feedback and constructive suggestions. We have revised our manuscript accordingly. In the revised manuscript, changes made in response to the reviews are indicated in blue. Line numbers are referenced in the responses below.

Reviewer 1

Comment: *line 52: It might be worthwhile to mention the mutant phenotype when SM migration is disrupted.*

Response: We have added a brief description of the phenotype here as suggested (lines 48-50).

ok

Comment: *Figure 1A: The detail in the “differentiation” worm schematic is difficult to see.*

Response: We have cropped and enlarged the middle region of the worm in this schematic.

Comment: *Figure 1B: UM and VM are hard to distinguish due to the size of the font, however, the colors of the cells help.*

Response: We enlarged the diagram and font size as much as possible within the constraints of the figure. We made the UM and VM labels in different colors to help further differentiate them.

Comment: *Figure 2: If possible, increase the size of the font used to label the cells.*

Response: We have increased the font size for the cell labels.

Comment: *line 234: It would be helpful to define “normal positioning.” Is this the precise center of the gonad (if so, how is this determined)? Or is there a small range relative to ventral cell positions?*

Response: We observed the normal position is for the SM to be centered on the dorsal uterine cell(s), dorsal to P6.p. This positioning is essentially invariable in wild-type animals. We defined normal positioning as having the SM cell body located over the uterine cells and P6.p. Scoring criteria are described in the methods, and we have added this definition to the results section where suggested (line 221).

Comment: *line 266: The authors state that there was the absence of a clear protein gradient observed, is it possible that there is a subtle gradient that is undetectable with the current technology/methods? If possible, please address this possibility.*

Response: Yes, it is possible that there is a subtle gradient that is undetectable with current methods. Indeed, a subtle gradient seems likely given that FGF acts instructively, but we are currently unable to determine to what extent an endogenous gradient exists in this context. We have added additional discussion of this possibility in the revised text (lines 261-263; 402-406). We do not think it would be possible to obtain better images using confocal microscopy as our spinning disk system is already optimized for sensitivity. The low expression levels and very small size of the extracellular space in the region where SMs migrate make fluorescence correlation spectroscopy infeasible to measure extracellular FGF abundance or diffusion parameters in this case.

Comment: *Figure S7: The fonts in both A and B are difficult to read, please consider increasing the font size. Statistical analyses would enhance the data interpretation.*

Response: We have increased the font size and added statistical analyses.

Comment: *line 513: Does the type of high-precision cover glass matter? If so, can details be provided?*

Response: The exact type of cover glass should not matter, but using high-precision glass means that the objective correction collar rarely needs to be adjusted for maximum brightness. We have added the supplier and catalog number in the methods section (line 496).

Comment: *line 554: Include criteria for identification as L3 and L4 stages.*

Response: We used a combination of somatic gonad development (dorsal uterine cell number), VPC number, and/or vulval morphogenesis to ensure that we scored SM position at a developmental stage when migration would normally be complete. We did not separate worms into L3 or L4 stages. The staging methodology is clarified in the revised methods (lines 545-547).

Reviewer 2

Comment: *Are cytonemes "seeable" in *C. elegans*? A reference of an example of visualizing *C. elegans* cytonemes would help. Otherwise, it is difficult to rule out cytonemes.*

Response: In *C. elegans*, we have observed dynamic filopodia in several types of non-migratory cells that resemble cytonemes observed in zebrafish and fly embryos. While we did not observe cytoneme-like structures during SM migration, we did observe numerous dynamic filopodia after SMs have finished migrating. These cytoneme-like structures are visible in post-migratory SMs using the same transgenes that we used for live imaging SM migration, which demonstrates this transgene is capable of visualizing cytoneme-like protrusions where they exist. We have added this point to the results and emphasized it in the discussion (lines 94-98; 347 - 350) along with a supplemental movie (Movie S1) showing cytoneme-like structures at a later developmental stage.

Comment: *In Figure 3G with ectopic *egl-17* expression, the cell is clearly not as polarized to the anterior as wild type. However, it is not polarized to the posterior, which is described in the text. It would be helpful to show a posteriorly polarized cell in this figure. demonstrating instructive reorientation by a diffusible FGF cue.*

Response: We have revised Fig. 3G to include another image showing a posteriorly migrating SM in addition to the mispolarized cell shown in the original submission. SMs migrating towards the tail in this strain do not have the same morphology as during normal migration, which we have clarified in the text (line 194-195).

Comment: *While this work clearly shows an instructive role for diffusible FGF, there is no evidence of any sort of functional gradient.*

Response: We agree that there is not evidence for a functional gradient, which is surprising considering that FGF/EGL-17 is diffusible, and FGFs are conventionally thought to form concentration gradients. While we cannot definitely determine whether a weak FGF gradient exists during SM migration using current technology, our data conclusively show that FGF/EGL-17 acts instructively in SM migration, and diffusion is required for SM migration, which are the primary conclusions. We have added additional consideration of the potential for a weak gradient in the revised text (lines 261-263; 402-406).

Comment: *p. 8 The authors mention "the cells are capable of measuring the relative strengths of opposing sources" of FGF. What is the basis for measuring relative strength of opposing sources to correlate with the phenotypic effect? I think the authors are implying that these cells can sense different amounts of FGF and respond differently. However, the cells might simply be attracted to the strongest source, and not comparing levels.*

Response: We apologize this was unclear in the text. Our data suggest that cells can migrate towards the strongest source of FGF, but do not show different types of responses to differing FGF levels. We have removed this inference from the revised text.

Reviewer 3:

Comment: *The authors stated that the endogenously tagged *EGL-17::mNG* is very faint. Thus it is possible that endogenous *EGL-17* has a graded distribution. However, the lower levels of *EGL-17* might be under the limit of detection. In fact, the image shown in Figure 4B using the *myo-2* promoter to drive *EGL-17::mNG* clearly showed a graded distribution of the *mNG* signal away from the pharynx where *EGL-17::mNG* is produced.*

Response: Yes, it is possible that there is an endogenous FGF gradient that is below the level of detection by confocal microscopy but still perceptible by the cells. We did observe a gradient distribution in the pharyngeal and tail misexpression experiments. However, much of the gradient in this misexpression strains appears to be within the hypodermal syncytium, with little protein

visibly associated with other cell types or the extracellular space. These promoters are also far stronger than endogenous FGF expression, and it is not clear how faithfully FGF dispersal in the misexpression experiments represents its native distribution processes. We have added additional consideration of these points (lines 261-263; 402-406).

Comment: *The expression of EGL-17 in the M lineage cells is intriguing. While the authors showed that expressing EGL-17 in the M lineage is not able to rescue the SM migration phenotype of egl-17 mutants, is M lineage expression of EGL-17 required for SM migration? In other words, does M lineage depletion of EGL-17 affect SM migration?*

Response: We are also intrigued by the early *egl-17* expression in the M lineage, but we have not been able to determine any function for this early expression. Several lines of evidence suggest early EGL-17 expression in the M lineage is not required for SM migration. First, our finding that SMs can migrate in the *Pmyo-2>egl-17* misexpression experiment demonstrates that early *egl-17* expression in the M lineage is not strictly essential for SM migration. We have noted this point in the revised manuscript (lines 227-229). While we are unable to deplete EGL-17 itself in the early M lineage, we did test roles for FGF signaling at different time points using timed, auxin-mediated depletion of endogenously tagged FGFR/EGL-15::mNG::AID in the M lineage. Continuous auxin treatment in *egl-15::mNG::AID; Phlh-8>TIR1* animals causes SM migration defects that phenocopy *egl-15(5a)* loss of function mutants (Gibney *et al.*, 2025, *bioRxiv*). However, we found that SMs migrate normally when larvae are treated with auxin after hatching followed by washout at the time when the SMs are born. This experiment indicates that early FGFR signaling in the M lineage is not required for later SM migration. We felt it was informative to note these results in our response to the review, but we elected not to include them in the revised manuscript because they are not central to our conclusions and are more relevant to another manuscript in preparation.

Comment: *It is not hard to imagine why the authors observed both normal migration as well as under migration and posterior migration when they expressed ELG-17 in the tail in WT animals. However, in Pmyo-2>FGF/EGL-17::mNG; FGF/egl-17(0) animals, a substantial number of animals have under-migrated SMs (Figure 4E). The authors should comment on this.*

Response: Our working hypothesis is that undermigration in a subset of cells could be caused by insufficient FGF. We hypothesize that pre-migratory SMs are located near the end of a potential FGF gradient emanating from the pharynx, and FGF levels are too low to support migration in some cells. We were unable to formally test this hypothesis but have added a comment on this possibility in the revised text (“The observation of undermigrated cells suggests that SMs may be born near the tail end of a potential FGF gradient in these animals where FGF levels may be near a lower limit required for migration guidance.” lines 222-224).

Comment: *Please comment on the internalized EGL-17::mNG in migrating SMs. Are these perdurance of EGL-17::mNG that was initially expressed in the M lineage cells before SMs were born, or are these internalized EGL-17::mNG taken up by the SMs along the migrating path?*

Response: In the revised manuscript, we used FRAP to assess the timing of EGL-17::mNG internalization in migrating SMs. We photobleached EGL-17::mNG in SMs at the onset of the FGF-dependent phase of migration, recovered animals to allow cells to migrate, and imaged again near the end of SM migration. EGL-17::mNG fluorescence in SMs recovered during the process of migration, which indicates that SMs do take up EGL-17::mNG along their migratory path. It is possible that some protein is also retained from earlier timepoints, but these experiments indicate a substantial fraction of the EGL-17::mNG visible in migrating cells is internalized during migration. These data are discussed on lines 245-250 and shown in Fig. 5G of the revised manuscript.

Other revisions not highlighted:

Minor rewording to make abstract under 180 words. Minor edits to the text to reduce word count for *Development* formatting.

Second decision letter

MS ID#: dev.204802R1

MS Title: FGF diffusion is required for directed migration of postembryonic muscle progenitors in *C. elegans*

Authors: Theresa V Gibney; Ariel M Pani
Article Type: Research Article

Dear Dr Pani,

I am happy to tell you that your manuscript has been accepted for publication in *Development*, pending our standard publication integrity checks.

Reviewer 1

I am satisfied with the revisions of the manuscript.

Reviewer 2

SUMMARY OF THE ADVANCE MADE IN THIS PAPER AND ITS POTENTIAL SIGNIFICANCE TO THE FIELD

The work shows that FGF diffusion is important for directed cell migration in vivo.

SUGGESTIONS TO AUTHORS

The authors have addressed reviewer comments.

Reviewer 3

SUMMARY OF THE ADVANCE MADE IN THIS PAPER AND ITS POTENTIAL SIGNIFICANCE TO THE FIELD

The revised manuscript adequately addressed previous reviewers' comments.